# Treatment of Basal Cell Carcinoma with Electrochemotherapy: Insights from the InspECT Registry (2008–2019)

Giulia Bertino [1], Tobian Muir [2], Joy Odili [3], Ales Groselj [4], Roberto Marconato [5], Pietro Curatolo [6], Erika Kis [7], Camilla Kjaer Lonkvist [8], James Clover [9,10], Pietro Quaglino [11], Christian Kunte [12,13], Romina Spina [14], Veronica Seccia [15], Francesca de Terlizzi [16], Luca Giovanni Campana [17,*] and the InspECT BCC Working Group [†]

1   Department of Otolaryngology-Head and Neck Surgery, IRCCS Policlinico San Matteo Foundation, 27100 Pavia, Italy; g.bertino@smatteo.pv.it
2   South Tees NHS Foundation Trust, Middlesbrough TS4 3BW, UK; tobian.muir@nhs.net
3   Department of Plastic Surgery, St. George's University Hospitals NHS Foundation Trust, London SW17 0QT, UK; joy.odili@stgeorges.nhs.uk
4   Department of Otorhinolaryngology and Cervicofacial Surgery, University Medical Centre Ljubljana, Zaloska 2, 1000 Ljubljana, Slovenia; ales.groselj@kclj.si
5   School of Surgery, University of Padova, 27100 Padova, Italy; roberto.marconato@iov.veneto.it
6   Dermatology Unit, Department of Internal Medicine and Medical Specialties, University "La Sapienza", 00042 Rome, Italy; pietro.curatolo@fondazione.uniroma1.it
7   Department of Dermatology and Allergology, Albert Szent-Györgyi Clinical Centre, University of Szeged, 6700 Szeged, Hungary; kis.erika.gabriella@med.u-szeged.hu
8   Department of Oncology, Herlev and Gentofte Hospital, University of Copenhagen, 2730 Herlev, Denmark; camilla.kjaer.loenkvist@regionh.dk
9   Cork Cancer Research Centre, University College Cork, T12 YN60 Cork, Ireland; j.clover@ucc.ie
10  Department of Plastic Surgery, Cork University Hospital, T12 DC4A Cork, Ireland
11  Department of Medical Sciences, Dermatologic Clinic, University of Turin, 10094 Turin, Italy; pietro.quaglino@unito.it
12  Department of Dermatologic Surgery and Dermatology, Artemed Fachklinik München, 81379 Munich, Germany; christian.kunte@artemed.de
13  Department of Dermatology and Allergology, Ludwig-Maximillian University, 80539 Munich, Germany
14  Psychology Unit, University Hospital of Padova, 35100 Padova, Italy; romina.spina@aopd.veneto.it
15  Otolaryngology, Audiology, and Phoniatric Operative Unit, Department of Surgical, Medical, Molecular Pathology, and Critical Care Medicine, Pisa University Hospital, Via Paradisa 2, 56124 Pisa, Italy; v.seccia@ao-pisa.toscana.it
16  IGEA Clinical Biophysics Department, Via Parmenide 10/A, Carpi, 41012 Modena, Italy; f.deterlizzi@igeamedical.com
17  Department of Surgery, Manchester University NHS Foundation Trust, Manchester Royal Infirmary, Oxford Rd, Manchester M13 9WL, UK
*   Correspondence: luca.campana@mft.nhs.uk
†   Membership of the InspECT BCC Working Group is provided in the Acknowledgments.

**Abstract:** This prospective registry-based study aims to describe electrochemotherapy (ECT) modalities in basal cell carcinoma (BCC) patients and evaluate its efficacy, safety, and predictive factors. The International Network for Sharing Practices of Electrochemotherapy (InspECT) multicentre database was queried for BCC cases treated with bleomycin-ECT between 2008 and 2019 (n = 330 patients from seven countries, with 623 BCCs [median number: 1/patient; range: 1–7; size: 13 mm, range: 5–350; 85% were primary, and 80% located in the head and neck]). The procedure was carried out under local anaesthesia in 68% of cases, with the adjunct of mild sedation in the remaining 32%. Of 300 evaluable patients, 242 (81%) achieved a complete response (CR) after a single ECT course. Treatment naïvety (odds ratio [OR] 0.35, 95% confidence interval [C.I.] 0.19–0.67, p = 0.001) and coverage of deep tumour margin with electric pulses (O.R. 5.55, 95% C.I. 1.37–21.69, p = 0.016) predicted CR, whereas previous radiation was inversely correlated (O.R. 0.25, p = 0.0051). Toxicity included skin ulceration (overall, 16%; G3, 1%) and hyperpigmentation (overall, 8.1%; G3, 2.5%). At a 17-month follow-up, 28 (9.3%) patients experienced local recurrence/progression. Despite no convincing evidence that ECT confers improved outcomes compared with standard surgical excision, it can still be considered an opportunity to avoid major resection in patients unsuitable for more

demanding treatment. Treatment naïvety and coverage of the deep margin predict tumour clearance and may inform current patient selection and management.

**Keywords:** basal cell carcinoma; skin cancer; bleomycin; electroporation; electrochemotherapy

## 1. Introduction

In its various forms, surgical resection is the mainstay of treatment for patients with basal cell carcinoma (BCC), the most common form of skin cancer. Generally, standard excision is indicated for small/intermediate-sized tumours in low-risk anatomical areas, whereas Mohs surgery is reserved for large or recurrent lesions, particularly in high-risk areas [1,2].

However, contemporary population ageing poses new therapeutic challenges, and patients with relevant comorbidities are best served by nonsurgical approaches, including radiation, photodynamic therapy, cryotherapy, and topical agents. Additionally, relatively few patients with locally advanced or metastatic BCC can now benefit from systemic treatment with hedgehog (i.e., vismodegib and sonidegib) and, more recently, programmed death-1 (i.e., cemiplimab) inhibitors [1,3].

In pursuing well-tolerated minimally invasive options, electrochemotherapy (ECT) was introduced in the late 1990s in the United States and Europe as a highly effective skin-directed locoregional chemotherapy. ECT harnesses short, high-voltage electric pulses to increase cell permeability to bleomycin or cisplatin, two otherwise poorly permeant drugs [4], and its mechanisms of action include the direct cytotoxic effect, composite antivascular activity, and a local immune response [5]. The procedure is currently widely used, mainly as a palliative measure, to provide local disease control in patients with superficial malignancies from various histotypes.

According to recent studies, ECT is associated with 50–100% clearance rates in BCC [6–8]. However, the generalizability of results is blurred by global variation in study designs, treated patients, and the reporting of results [9–20]. The present study aimed to describe ECT modalities in the largest available cohort of BCC patients, evaluate treatment efficacy and safety, and identify predictors of response. Its results are clinically relevant because they may inform the current patient selection and help design future clinical trials in more selected populations.

## 2. Materials and Methods

### 2.1. Study Design

We analysed the International Network for Sharing Practices of ECT (InspECT) database. The InspECT registry is a prospective database established in 2008 to assess the outcome of patients with skin cancers or cutaneous metastases treated with ECT (https://insp-ect.eu accessed on: 20 May 2022), approved by the National Medical Ethics Committee of the Republic of Slovenia (No. 102/09/14). The study was carried out according to the rules of Good Clinical Practice and tenets of the Declaration of Helsinki. The collected information included patient demographics, tumour characteristics, ECT parameters (anaesthesia regimen, chemotherapy drug, type of pulse applicator, duration of the procedure), tumour response, treatment toxicity, and patient-reported outcomes. The primary endpoints were activity and safety; secondary endpoints were identifying predictors of response and assessment of tumour control.

### 2.2. Participants

The InspECT database was queried for all BCC patients (any histological subtype) treated between 2008 and 2019. A multidisciplinary team agreed upon the indication to ECT; each participant received information regarding the treatment intent and then signed a consent form. The inclusion and exclusion criteria were derived from the European

Standard Operative Procedures of ECT (ESOPE) [21,22]. In particular, the candidate patients had to have primary or recurrent BCC in which other treatment modalities had failed or were impracticable. In addition, normal lung and renal function were prerequisites to being suitable for the procedure. Absolute contraindications to the treatment were allergy to bleomycin or a previous cumulative dose of 400,000 IU/m$^2$ due to the risk of lung fibrosis, pregnancy, or lactation [21,22]. Preoperative workup included a complete medical history, physical examination, and standard blood tests.

### 2.3. Procedure

2.3.1. Protocol

The procedure was initially performed following the 2006 ESOPE guidelines and, from 2018, their updated version [21,22], with no changes in chemotherapy doses or electroporation protocol.

2.3.2. Anaesthesia

Local anaesthesia, eventually associated with mild sedation, was used in patients with small (<3 cm) and few ($n \leq 7$) lesions, whereas general sedation/anaesthesia was added in patients with tumours larger than 3 cm, mainly when located (a) close to the periosteum, (b) in sensitive regions (e.g., chin, cheek, lip), or (c) in patients needing airway protection.

2.3.3. Chemotherapy

Bleomycin was administered intravenously or intratumourally, depending on tumour features and patient preoperative assessment. The intravenous bolus was administered at 15,000 IU/m$^2$ of body surface area, with a maximum dose capped at 30,000 IU. Dose adjustment for age or impaired renal function was used according to local guidelines. Bleomycin infusion lasted 2–5 min, and the electric pulses were applied after 8 min when the drug had diffused into tissues. As for the intratumoural route, the recommended concentration of bleomycin solution was 1000 IU/ml, and the injected volume was determined according to the lesion volume.

2.3.4. Electric Pulse Delivery

Electrode choice was based on BCC location, size, and morphology and included needle or plate electrodes of different sizes and geometries. The electric fields (eight pulses of 100 ms duration, with an amplitude of 1000 V/cm for needle or 1300 V/cm for plate electrodes) were delivered through a pulse generator certified for clinical application (Cliniporator$^{TM}$, IGEA, Carpi, Italy). Intraoperatively, the timing of pulse delivery varied according to the route of bleomycin administration (immediately after intralesional injection, after 8 min following intravenous infusion).

2.3.5. Postprocedural Care

The treated lesions were covered with nonadherent dressings or alginates in cases of ulceration. Patients treated under general anaesthesia were monitored for 12/24 h.

### 2.4. Outcome Assessment

For each patient, the largest tumour diameter was measured using a millimetre ruler at baseline and one- and two-month follow-up, following the Response Evaluation Criteria in Solid Tumours (RECIST v1.1). Histological verification was not routinely performed but allowed according to clinical judgment. Complete response (CR) was defined as tumour disappearance, whereas partial response (PR) was a shrinkage of at least 30%, and progressive disease (PD) was an increase of at least 20%. The cases where inflammation or ulceration hampered response assessment were labelled as "not evaluable" (NE). Responses defined neither by PR nor PD criteria were considered stable disease (SD). Pain intensity was evaluated through a numerical rating scale (NRS) ranging from 0 to 10 (no pain and maximum pain, respectively). Toxicity and complications were graded from grade 1 (G1,

mild adverse event) to grade 4 (G4, life-threatening/disabling adverse event) according to the National Cancer Institute (NCI) Common Terminology Criteria for Adverse Events (CTCAE) v3.0. Except for the two-month visit to assess the response, as per RECIST, patient follow-up was in accordance with local institutional protocols. Local recurrence within the ECT field was diagnosed based on clinical and histological findings, as appropriate.

### 2.5. Statistical Analysis

Univariate response analysis was performed with contingency tables of frequency and Pearson's chi-squared test with Yates' continuity correction. Multivariate analysis of variance (MANOVA) was used to identify predictors of response. Local progression-free survival (LPFS) was calculated from the first ECT to the date of relapse or local progression or last follow-up. The survival curve was calculated by the Kaplan–Meier method. Pain scores were compared using coupled Student's $t$-test analysis. $p$-value < 0.05 was considered statistically significant. Analyses were performed with NCSS 9 software (NCSS, LLC. Kaysville, UT, USA).

## 3. Results

### 3.1. Patient Population

A total of 330 patients were consecutively treated at 15 European centres (Table 1); 30 subjects were excluded from the response analysis due to one of the following reasons: the inability or unwillingness to attend follow-up ($n = 14$), application of other treatments ($n = 6$), death from other cause ($n = 5$), or lost to follow-up ($n = 5$). As a result, 300 patients (188 males, 112 females) with 587 tumours were evaluable. The treatment intent was palliative in 39 cases (13%). The median BCC size was 13 mm (range 5–350 mm), and 90% of tumours (560/587) were smaller than 3 cm. Most (80%) were located in the head and neck, and 5% were ulcerated.

**Table 1.** Patient and treatment characteristics (330 patients with 623 tumours).

| Characteristic | No. (%) or Median (Range) |
|:---:|:---:|
| Sex | |
| Males | 205 (62%) |
| Females | 125 (38%) |
| Age (years) | 76 (23–98) |
| No. tumours/patient | 1 (1–7) |
| Tumour presentation | |
| Primary naïve | 200 (61%) |
| Primary persistent | 79 (24%) |
| Recurrent | 51 (15%) |
| Tumour size (mm) | 13 (5–350) |
| ≤3 cm | 560 (90%) |
| >3 cm | 63 (10%) |
| Anatomical location | |
| Head and Neck | 496 (80%) |
| Trunk | 74 (12%) |
| Limbs | 53 (8%) |
| Previous treatment | |
| None | 200 (61%) |
| Surgical excision | 116 (35%) |
| Other [a] | 12 (3%) |
| Unknown | 2 (1%) |
| Anaesthesia | |
| Local | 223 (68%) |
| Local + sedation | 107 (32%) |
| BLM administration | |

**Table 1.** *Cont.*

| Characteristic | No. (%) or Median (Range) |
|---|---|
| Intratumoural | 146 (44%) |
| Intravenous | 184 (56%) |
| Electrode geometry | |
| Row needle | 383 (61.5%) |
| Hexagonal needle | 164 (26.3%) |
| Plate | 44 (7.1%) |
| Combination | 23 (5.1%) |
| Retreatment (No. of pts) | 52 (16%) |
| Interval to 2nd ECT (months) | 6.7 (1.2–47) |

[a] radiotherapy, $n = 4$; cryotherapy, $n = 2$; photodynamic therapy, $n = 2$; topical therapies, $n = 3$; ECT, $n = 1$.

### 3.2. Treatment

The procedure was performed under local anaesthesia in 67% of cases (Table 1). Bleomycin was administered intravenously or intratumourally in 57% and 43% of patients.

### 3.3. Toxicity

Adverse events included skin hyperpigmentation (7%, $n = 22$ patients; 2 with grade-2) and skin ulceration (6.7%, $n = 20$; 3 with grade-3). Other uncommon side effects included local oedema ($n = 6$), flu-like symptoms ($n = 4$), rash/allergic reaction ($n = 3$), skin atrophy ($n = 1$), hypopigmentation ($n = 1$). Pain scores increased in the immediate postprocedural assessment, although 96% of patients (96%) experienced no/mild discomfort. Local pain scores assessed at two months and the last follow-up (median 12 months [range, 2.3–78]) were significantly lower compared with baseline (Table 2). Finally, the proportion of patients reporting the consumption of analgesics decreased from 9% at baseline to 5% at the last follow-up ($p = 0.04$).

**Table 2.** Assessment of pain ($n = 300$ patients).

| Score [a] | Baseline | Post-Procedure | Two Months | Last Follow-Up [b] |
|---|---|---|---|---|
| Median (range) | 0 (0–7) | 0 (0–8) | 0 (0–10) | 0 (0–9) |
| Mean $\pm$ SD | 0.41 $\pm$ 1.11 | 0.49 $\pm$ 1.14 | 0.32 $\pm$ 1.27 | 0.23 $\pm$ 1.11 |
| T-test vs. baseline | | $p = 0.017$ | $p = 0.347$ | $p = 0.023$ |

[a] Numerical rating score ranging from 0 (no pain) to 10 (maximum pain). [b] At a median of 12 months (range 2.5–78).

### 3.4. Tumour Response

Results are presented in Table 3. Following the first ECT course, the per-tumour clearance rate was 83% (488/587 lesions). Out of the 300 evaluable patients, 242 (81%) achieved CR, 46 (15%) PR, 9 (3%) SD, whereas 3 (1%) were NE. Two representative cases are shown in Figures 1 and S1.

**Table 3.** Tumour response following the first ECT application ($n = 300$ patients with 587 tumours).

| Response | Per-Patient | | Per-Tumour | |
|---|---|---|---|---|
| | *n* | % | *n* | % |
| CR | 242 | 80.7 | 488 | 83.1 |
| PR | 46 | 15.3 | 76 | 12.9 |
| SD | 9 | 3.0 | 15 | 2.6 |
| PD | 0 | 0 | 0 | 0 |
| NE | 3 | 1.0 | 8 | 1.4 |
| Total | 300 | 100 | 587 | 100 |

Note: CR, complete response; PR, partial response; SD, stable disease; PD, progressive disease; NE, not evaluable.

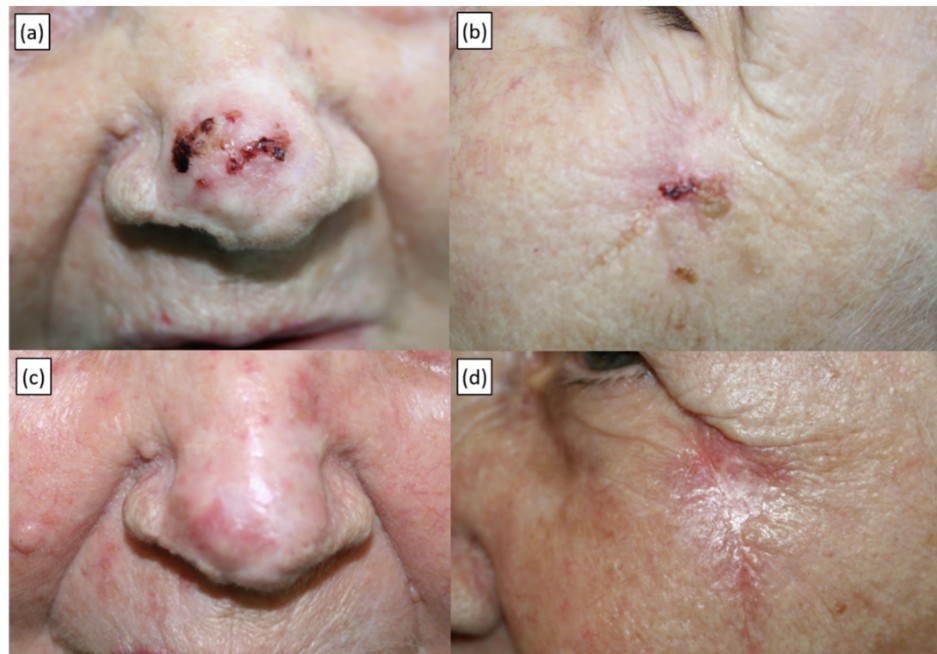

**Figure 1.** Recurrent BCC of the tip of the nose and concomitant primary BCC of the left zygomatic area in an 89-year-old patient treated with a single ECT application under local anaesthesia. (**a,b**) Baseline presentation. (**c,d**) Twelve-month follow-up showing a complete response.

### 3.5. Predictors of Response

By univariate analysis, small ($\leq$3 cm) tumour size ($p = 0.002$), absence of previous treatments ($p < 0.001$), no prior radiotherapy ($p = 0.007$), head and neck location ($p = 0.008$), and the coverage of deep ($p = 0.001$) and lateral margins ($p = 0.045$) were associated with tumour clearance, whereas only the coverage of deep tumour margin (odds ratio [O.R.] 5.44, $p = 0.016$) and the absence of previous treatments (O.R. 2.86, $p = 0.001$) remained independent predictors in multivariate analysis (Table 4). In multivariate analysis for the overall response, the significant variables were the absence of prior treatments, small tumour size, and head and neck location (Table S1).

**Table 4.** Predictors of complete response.

| Variable | Univariate | | | Multivariate | | |
|---|---|---|---|---|---|---|
| | O.R. | 95% C.I. | *p* | O.R. | 95% C.I. | *p* |
| Anatomical location (head/neck vs. other) | 2.75 | 1.30–5.83 | 0.008 | 1.98 | 0.86–4.52 | 0.107 |
| Route of BLM (i.v. vs. i.t.) | 0.60 | 0.33–1.09 | 0.095 | | | |
| Previous RT (yes vs. no) | 0.18 | 0.05–0.62 | 0.007 | 0.25 | 0.06–1.00 | 0.051 |
| Lymphedema (yes vs. no) | 0.24 | 0.01–3.84 | 0.311 | | | |
| Tumour size (< vs. $\geq$30 mm) | 2.48 | 1.15–5.35 | 0.020 | 1.68 | 0.71–3.99 | 0.237 |
| Coverage of deep margins (yes vs. no) | 3.46 | 1.79–7.06 | 0.001 | 5.44 | 1.37–21.69 | 0.016 |
| Coverage of lateral margins (yes vs. no) | 2.23 | 1.02–4.88 | 0.045 | 0.37 | 0.08–1.61 | 0.185 |
| Presentation (recurrent vs. primary) | 0.50 | 0.25–1.00 | 0.051 | | | |
| No previous treatments | 3.45 | 1.89–6.25 | <0.001 | 2.86 | 1.49–5.26 | 0.001 |

Note: O.R., odds ratio; C.I., confidence interval.

### 3.6. Local Control and Patient Survival

Fifty-two patients (17%) underwent a second ECT after a median of 7 months (range, 1–47 months) for PR ($n = 37$) or recurrence ($n = 15$). With a median follow-up of 17 months

(mean 22, range 2–103), 28 (9.3%) patients experienced local progression after a median interval of 13 months (range 3–46). Local progression included relapse after a previous CR (*n* = 26 patients) or progression after a previous PR (*n* = 27 patients). No significant characteristics emerged from the analysis of patients with recurrence (Table S2). Of these 28 subjects, 13 received a second ECT and 9 (69%) achieved CR. Overall, 1- and 2-year local progression-free survival (LPFS) was 96% (95% C.I. 93–98%) and 90% (95% C.I. 86–94%), respectively (Figure 2). Fifteen patients (5%) died of other causes after a median of 14 months (range 2–28 months).

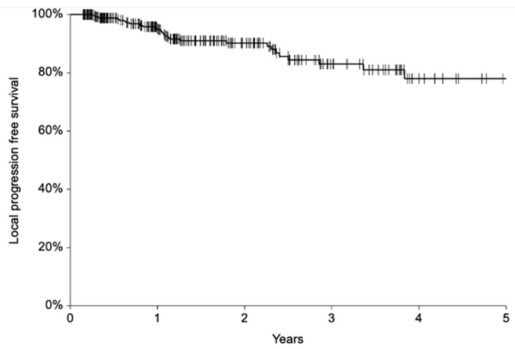

**Figure 2.** Local progression-free survival.

## 4. Discussion

### 4.1. ECT Safety

The present registry-based study relies on the largest BCC series treated with ECT thus far and provides new critical insights into the application of this alternative nonsurgical therapy. Importantly, our findings reinforce the notion that ECT is a safe, although suboptimal, therapeutic option for BCC patients. Most participants reported no adverse events, and local toxicity ranged to 7%, mostly mild, consistent with outcomes reported with other skin-directed therapies [23]. In this regard, it is worth noting that preliminary evidence suggests that de-escalated doses of BLM or the use of calcium instead of chemotherapy may further reduce risk or even avert toxicity while preserving ECT efficacy [17].

### 4.2. Predictors of Response

Based on our results, ECT efficacy seems to be influenced by biological and technical factors. More than 80% of patients achieved a cure following a single course of treatment, and, interestingly, CR achievement correlated with BCC naïvety and the coverage of a deep tumour margin with electric pulses. These findings confirm the observations from a retrospective study in patients with head and neck cancers, where ECT demonstrated higher efficacy in chemo-naïve tumours [20]. In these patients, the absence of clone selection and preserved vasculature in radiotherapy-naïve tumours likely leads to unperturbed chemotherapy distribution and higher ECT efficacy.

As for the technical aspects, it should be acknowledged that electrode placement is operator-dependent and generally based only on clinical judgement; as such, uncertainty exists as to the depth and extension of electrode placement, also depending on BCC histological subtypes (Figure 3). This variation in treatment application needs to be addressed in future studies. Meanwhile, the predictive factors identified in this study can be helpful in clinical practice to select ECT candidates (i.e., patients with treatment-naïve BCC with tumours suitable for adequate electrode application) and inform clinicians regarding the need for retreatment based on intraoperative feedback (i.e., adequacy of electric fields on tumour margins). Additionally, they can support researchers in designing future studies in well-defined cohorts of patients.

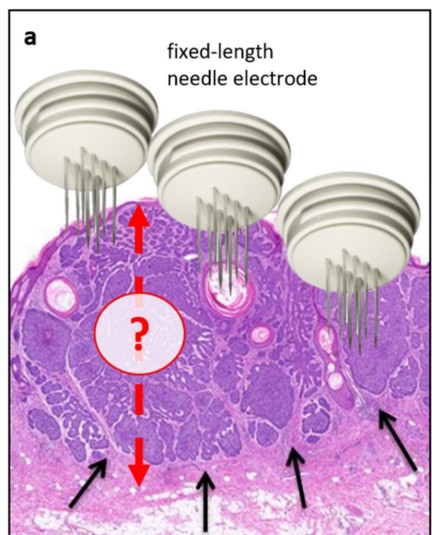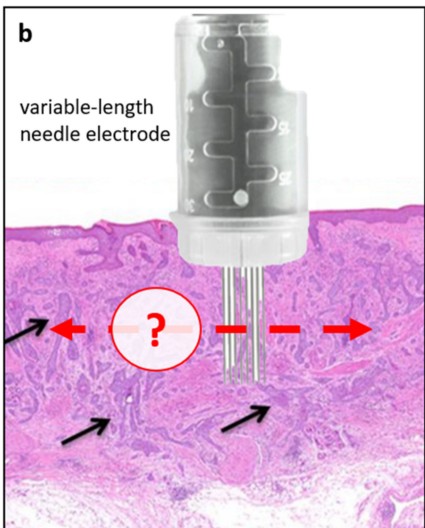

**Figure 3.** Intraprocedural challenges of ECT in BCC. As a principle, effective ECT treatment delivery relies on adequate chemotherapy distribution within the tumour (following i.v. or i.t. administration) and complete coverage with electric fields (electroporation). In this regard, lower response rates are associated with previous radiotherapy (likely due to its effect on microcirculation) and the insufficient electroporation of tumour margins, an aspect closely associated with the modalities of electrode application. Generally, the extent and depth of electrode insertion are operator-dependent and based on clinical judgment. (**a**) An example of nodular BCC with well-circumscribed areas of cancer cells (black arrows) treated with a fixed-length electrode. In this case, the actual BCC depth may be underestimated, thus leaving deep portions of the tumour untreated. (**b**) An example of infiltrative BCC in which the actual extension of the tumour may be challenging to discriminate with potential repercussions on the coverage of lateral tumour margins. Red arrows indicate the potential depth (**a**) or extension (**b**) of electrode application.

In contrast to previous publications, the present analysis did not find that tumour size correlates with the response to ECT [8,18]. However, it should be noted that 90% of BCCs were <3 cm. Nonetheless, tumour size and head and neck location proved to be significant predictors of overall response (Table S1) and may still help refine patient selection and clinical decision-making in palliative cases.

### 4.3. Local Control

BCC is a slowly growing tumour; therefore, long-term patient outcome is essential in evaluating treatment efficacy. Critically, the recurrence rate at two years was 10%. This result is a suboptimal outcome compared with conventional treatments and necessitates the extended follow-up of our patients. The five-year recurrent rate in primary BCC is 1–3% following Mohs surgery and 8–10% following excision or radiotherapy. In recurrent BCC, 5-year recurrence rates are 5.6%, 17.4%, and 9.8%, respectively. However, the heterogeneity of our cohort (which included both primary and recurrent BCC, a small (10%) but not negligible proportion of tumours >3 cm, and 13% of patients treated with palliative intent due to frailty or comorbidities) makes it difficult to perform rigorous comparisons with other treatments.

Surprisingly, the variables associated with tumour clearance in multivariate analysis were not associated with local control (Table S2). Hence, we speculate that other factors may determine the sensitivity to bleomycin and ultimately dictate treatment outcomes in the intermediate term. These include (a) the low growth kinetics of BCC (bleomycin is selectively toxic to cells in the M and G2 phases of the cell cycle), (b) the fraction of hypoxic cells (oxygen is an essential substrate for bleomycin's action), and (c) other biological determinants of bleomycin cytotoxicity (e.g., metabolic inactivation by bleomycin hydrolase, the integrity of the DNA repair systems). In this regard, the InspECT group is

committed to investigating the biological determinants of response to ECT and has recently proposed a roadmap to pursue this goal [24].

### 4.4. Previous Clinical Experiences

Few studies have investigated ECT in BCC, broadly varying in design and patient characteristics (Table 5).

### 4.5. Comparison with Other Treatments

Thus far, ECT has not been formally compared against other skin-directed nonsurgical therapies. However, the results from a 2014 meta-analysis in patients with cutaneous metastases (mainly melanoma and breast cancer) indicate similar efficacy and safety [23]. A recent trial randomised 99 patients with primary BCC between ECT with intratumoural bleomycin and standard surgical excision. At three years, the clearance rate was comparable (87.5% and 97.5%, respectively) [7]. Only two studies reported long-term outcomes, with 5-year local control rates of 70% and 87% [7,8]. Finally, some mixed series enrolled variable proportions of BCC patients (Table S3). Among these, the IMI-GIDO ($n = 24$ BCC patients), EURECA ($n = 34$), and InspECT ($n = 298$) study, which reported clearance rates of 67%, 91%, and 71%, respectively [7,18,19].

### 4.6. The Role of ECT in Locally Advanced BCC

Interestingly, ECT was also effective in cases with BCC larger than 3 cm, of which a representative patient is presented in Supplementary Figure S1. Advanced disease may be a novel investigational area of research for ECT users. Although relatively rare, locally advanced BCC is a therapeutic challenge, and radical surgical resection may result in substantial deformity or morbidity. In this context, oral hedgehog inhibitors are the standard of care, having demonstrated remarkable efficacy, with response rates ranging from 57% to 69% in the most recent pooled analyses [25]. Nonetheless, toxicity (e.g., muscle spasms, dysgeusia, alopecia, and gastrointestinal symptoms) often imposes adjustments to the treatment schedule [1]. Interestingly, although preliminary, the favourable results reported with ECT in locally advanced BCC provide the rationale for investigating combined strategies with systemic treatment, where local therapy may help consolidate response, manage recurrences, or maintain local control during 'drug holidays' [8,18].

### 4.7. The Role of ECT in an Ageing Population

Nowadays, an increasing proportion of BCC patients are part in the elderly population, where performance status, comorbidities and BCC multifocality may prevent demanding treatments. On this note, the patients included in the present study were deemed unfit for more invasive options, and 13% were treated with palliative intent, as reflected by their short survival. Interestingly, ECT is a well-tolerated and highly effective option, even in the oldest-old population [26].

**Table 5.** Published series on ECT in BCC (additional studies in unselected populations are available in Table S3).

| Author | Year | Study | No Pts | No of Tumours | BCC Subtype | BCC Presentation | T Size (mm) | Drug | Route | CR (%) | Re-ECT (%) | F-Up (mos) | Recurrence |
|---|---|---|---|---|---|---|---|---|---|---|---|---|---|
| Clover [7] | 2020 | Randomised (ECT vs. surgery) | 52 | 69 | Nodular/Superficial/ Infiltrative/Morpheaform | Primary | 1.7 cm$^2$ (0.2–10) | BLM | i.t. | 86 [a] | 12 | 60 | 5-year LDFS, 87% (5 recurrences) |
| Kis [9] | 2019 | Case series | 12 | 17 | n.r. | Primary: 3 Recurrent: 9 | 11 (3–43) | BLM | i.v./i.t. | 100 | 33 | 19 | n.r. |
| Campana [10] | 2017 | Case series | 84 | 185 | Superficial/Nodular/ Infiltrative/Morpheaform | Primary/recurrent (L/LA/Mts) | 20 (5–267) | BLM | i.v./i.t. | 50 [a] | 29 | 49 | 5-year LPFS, 70% |
| Ruggeri [10] | 2015 | Case report | 1 | 3 | n.r | Recurrent, multifocal | 4, 7, 8 | BLM | i.v. | 100 | 0 | 7 | no |
| Salwa [11] | 2014 | Case series | 3 | 3 | n.r. | Primary, periocular | 0.5–1 cm$^2$ | BLM | i.t. | 100 | 0 | 5–8 | no |
| Gatti [12] | 2014 | Case report | 1 | 1 | n.r. | Recurrent | n.r. | BLM | i.v. | 100 | 0 | 12 | no |
| Kis [13] | 2012 | Case series | 3 [b] | 99 | Superficial/Nodular/ Ulcerated/Plaque | primary/recurrent | 9 (3–22) | BLM | i.v. | 87 [a] | 33 | n.r. | n.r. |
| Fantini[14] | 2008 | Case report | 1 | 3/3/ "multiple" | BCC with SCC differentiation | Metastatic | n.r. | BLM | i.t./i.v. | 100 | 0 | 8 | no |
| Glass [15] | 1997 | Case series | 20 | 54 | Nodular | Primary | 9 (4–21) | BLM | i.t. | 98 [c] | 10 | 18 | no |
| Glass [16] | 1996 | Case report | 2 | 6 | Superficial/ Nodular | n.r. | n.r. | BLM | i.v. | 33 [c] | 0 | n.r. | n.r. |

Note: BLM, bleomycin; CDDP, cisplatin; CR, clearance rate; ECT, electrochemotherapy; i.t., intratumoural; i.v., intravenous; LDFS, local disease-free survival; LPFS, local progression-free survival; mos, months; n.e., not evaluable; n.r., not reported; SCC, squamous cell carcinoma. [a] Per-tumour assessment; [b] All patients had Gorlin–Goltz syndrome; [c] Per-patient assessment.

*4.8. Study Limitations*

This report has limitations. The first is the single-arm design. The second is a selection bias due to a lack of data from patients with worse outcomes. Third, the heterogeneity of patients limits the generalisation of findings. Another drawback is the lack of stratification according to BCC histological subtypes because aggressive histology has been reported to affect ECT efficacy [8]. Furthermore, no biopsy was performed for response verification nor the assessment of patient-rated cosmetic outcomes. Notwithstanding, we believe that a subset of well-selected subjects benefits from ECT. This seems to be a safe alternative when standard treatment is not practicable due to patient refusal/comorbidities or tumour characteristics (location, size, or multifocality) [8,26]. Notably, the ECT procedure is standardised and straightforward; unlike radiotherapy, outcomes can be compensated by salvage treatments. In the era of minimally invasive surgery, this treatment could be an ideal choice in the effort to minimise the extent of surgical intervention. At the same time, as reported by the National Institute for Health and Care Excellence (NICE) and a recent meta-analysis, the current level of evidence remains low in quantity and quality [27,28], and the role of ECT in current guidelines is marginal [29]. Hence, there is a need for standardisation and continuous rigorous evaluation through comprehensive data collection and the monitoring of indications, outcomes, and costs [30,31]. Additionally, the understaging of the disease is a potential risk with ECT because of the absence of a surgical specimen in most procedures; thus, judicious application and postoperative monitoring are mandatory. Finally, the cost to the health systems of treating BCC, mainly when located on the face, is high; multidisciplinary teams need to be aware of an increasing number of options, their results, and their cost [32].

## 5. Conclusions

Based on the illustrated data, the proposed ECT indications can be summarised as follows: (1) treatment of low-risk BCC in strictly selected subgroups (patients with multiple BCC (e.g., Gorlin–Goltz syndrome), unfit for surgical treatment because of their age or comorbidities; (2) organ-preserving treatment of small-sized high-risk BCC located in delicate anatomical areas (e.g., eyelid, nose, and auricle) in patients unfit/unwilling surgical treatment to preserve patient functioning (in this subgroup, however, recurrent status or aggressive histotype deserves a note of caution); (3) ancillary treatment of locally advanced BCC in conjunction with systemic treatment (treatment of patients with near-complete response, local recurrence, patients needing "drug holidays" from systemic treatment because of intolerance/toxicity).

Many clinical challenges and questions remain. Moving forward, we envision the following line of investigation and possible improvements: (1) de-escalation of the bleomycin dose to reduce dermatologic toxicity and consolidate tolerability [17]; (2) use of calcium instead of bleomycin (based on positive accumulating evidence on the efficacy and safety of calcium electroporation) [33]; (3) identification of the best strategies for ECT use in combination with immunotherapy or targeted therapy; (4) assessment of the quality of life and aesthetic outcomes.

In conclusion, there is no convincing evidence to indicate that ECT is associated with better outcomes when compared with standard treatment in BCC. Therefore, patients with localised disease should undergo excisional surgery whenever feasible. However, when excision is not viable, ECT is a safe and reasonably effective alternative with a higher chance of success in treatment-naïve individuals whose tumours are entirely covered with electric pulses. These predictive factors may help ECT users refine patient selection and design future studies. International databases and investigation of biological determinants of response will be mandatory to further advances.

**Supplementary Materials:** The following supporting information can be downloaded at: https://www.mdpi.com/article/10.3390/curroncol29080423/s1, Figure S1: Primary locally advanced BCC in a 96-year-old patient treated with ECT under general anaesthesia; Table S1: Predictors of objective response;

Table S2: Patient characteristics according to development or not of local recurrence; Table S3: Clinical studies on ECT in unselected cancer populations, including patients with BCC [34–37].

**Author Contributions:** Conceptualisation, G.B., T.M. and A.G.; Patient treatment and follow-up, G.B., T.M., J.O., A.G., R.M., P.C., E.K., C.K.L., J.C., P.Q., C.K., V.S. and L.G.C.; methodology, F.d.T., R.S. and L.G.C.; formal analysis, G.B., T.M. and F.d.T.; data curation, F.d.T. and R.S.; writing—original draft preparation, G.B., T.M. and J.O.; writing—review and editing, A.G., R.M. and L.G.C. All authors have read and agreed to the published version of the manuscript.

**Funding:** This research received no external funding.

**Institutional Review Board Statement:** This study was conducted in accordance with the Declaration of Helsinki and approved by the National Medical Ethics Committee of the Republic of Slovenia (Protocol No. 102/09/14).

**Informed Consent Statement:** Written informed consent was obtained from the patient to publish this paper.

**Data Availability Statement:** The dataset generated during the current study is available from the corresponding author upon reasonable request.

**Acknowledgments:** InspECT BCC Working Group: Silvia Di Felice (University of Pavia, Italy), Graeme Moir (Barts Health NHS Trust, London, U.K.), Antonio Orlando (North Bristol NHS Trust, Bristol, U.K.), and Rowan Pritchard-Jones (Whiston Hospital, Prescot, U.K.). The authors would like to thank Alberto Tonelli for assisting with patient management. In the preparation of this manuscript, the authors followed the S-PRINT (Sustainable Paperless Reference Initiative Nourishes Trees) recommendation, printing 3 of 27 papers on the reference list (S-PRINT score: 11%) (https://twitter.com/LucaCampana_611/status/1140741813277483008 accessed on: 20 May 2022).

**Conflicts of Interest:** The InspECT database is hosted by IGEA S.p.A. (Carpi, MO, Italy) but administered by an independent board. FdT is an IGEA employee. GB, TM, AG, PC, RC, EK, AJPC, GM, JO, PQ, AO, CK, and LGC have received previous travel grants from IGEA unrelated to the present study. SDF, RM, CKL, and SHL have no conflict of interest to declare.

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
