# Peer review of "Treatment of Basal Cell Carcinoma with Electrochemotherapy: Insights from the InspECT Registry (2008–2019)"

_curroncol, doi:10.3390/curroncol29080423_

Round 1
Reviewer 1 Report
As there are few reviews on the clinical effects of ECT, it is a necessary study. The authors presented the strengths and deficiencies of ECT along with statistical analysis through the registry data. It is well organized, so it is recommended to publish it after a minor review in Current Oncology.
A high recurrence rate within 2 years seems to be the most characteristic feature of ECT, so a detailed analysis seems to be necessary. I wonder if there is any more information to be analyzed.
Notes (a)-(d) inside Figure 1 seems too small.
Some abbreviated words need full name, though they are generally used. Such as R.R. , OR, F-up (mos).
The notes for Legend in Table 5 should be checked again.
Table 5 is not easy to read due to line breaks.
Author Response
Q1: A high recurrence rate within 2 years seems to be the most characteristic feature of ECT, so a detailed analysis seems to be necessary. I wonder if there is any more information to be analysed.
R: Thanks for this question and the time to review the manuscript. To shed light on the recurrence rate, we analysed the characteristics of the patients who experienced tumour regrowth (results provided in Supplementary Table 2). We did not find patient, tumour or treatment variables associated with local recurrence.
Surprisingly, the variables associated with tumour clearance in multivariate analysis (no previous treatments, coverage of deep tumour margin) were not associated with local control during follow-up. [coverage of deep tumour margin is now included in v.R1 of the S-Table 2]
Hence, we speculate that other factors determine sensitivity to bleomycin and ultimately dictate treatment outcomes in the intermediate term. These include (a) the low growth kinetics of BCC (bleomycin is selectively toxic to cells in the M and G2 phases of the cell cycle), (b) the fraction of hypoxic cells (oxygen is an essential substrate for bleomycin's action), and (c) other biological determinants of bleomycin cytotoxicity (e.g. metabolic inactivation by bleomycin hydrolase, the integrity of the DNA repair systems).
In this regard, the InspECT group is committed to investigating the biological determinants of response to ECT (Sersa et al. Biological factors of the tumour response to electrochemotherapy: Review of the evidence and a research roadmap. EJSO 2021).
We have introduced these considerations in paragraph 4.3
---
Q2: Notes (a)-(d) inside Figure 1 seem too small.
R: Thanks for this suggestion. Figure 1 has been improved and is now provided as a high-resolution file.
---
Q3: Some abbreviated words need full name, though they are generally used. Such as R.R. , OR, F-up (mos).
R: The abstract, text and tables have been carefully checked, and all abbreviations provided in extenso.
---
Q4: The notes for Legend in Table 5 should be checked again.
R: Thanks for spotting this; we appreciate the opportunity to improve the manuscript. Table 5 legend has been reviewed, and the abbreviations and notes amended.
---
Q5: Table 5 is not easy to read due to line breaks.
R: Thanks for this request. Table 5 is now provided on a horizontal page to improve the spacing between columns and the readability of contents.
Reviewer 2 Report
The article is well written, the data are convincing and the limitations of the study are well represented. I only suggest to better describe in the discussion and in the conclusions sections the actual role of ECT in clinical settings and the potentiality of this application in the near future.
The main purpose of the article is to illustrate the data on the efficacy of ECT in the treatment of basal cell carcinomas, retrospectively analyzing data from the InspECT Registry (2008-2019).
The topic is interesting because it represents the largest clinical study published to date on the use of ECT for the treatment of cutaneous tumors.
As mentioned before, it is the largest and most homogenous study on the subject published to date
The authors to better specify the potential applications of this method in the clinic in the light of the data illustrated and the possible improvements to be made.
Author Response
Q1: The authors to better specify the potential applications of this method in the clinic in the light of the data illustrated and the possible improvements to be made.
R: Thanks for raising this question. The current potential ECT indications could be summarised as follows:
- treatment of low-risk BCC in well-selected subgroups (patients with multiple BCC [e.g. Gorlin-Goltz syndrome], unfit for surgical treatment because of their age or comorbidities
- treatment of small-size high-risk BCC located in delicate anatomical areas (eyelid, nose, auricle) in patients unfit/unwilling surgical treatment to preserve patient functioning
- ancillary treatment of locally-advanced BCC in conjunction with systemic treatment (treatment of patients with near-complete response, recurrence, patient needing "drug holidays" from systemic treatment because of intolerance/toxicity)
We envision the following improvements:
- de-escalation of the bleomycin dose to reduce dermatologic toxicity
- use of calcium instead of bleomycin (based on positive accumulating evidence on calcium electroporation)
- use in combination with immunotherapy or targeted therapy
These considerations are now included in the conclusions paragraph.